# Pretreatment Techniques and Green Extraction Technologies for Agar from *Gracilaria lemaneiformis*

**DOI:** 10.3390/md19110617

**Published:** 2021-10-30

**Authors:** Qiong Xiao, Xinyi Wang, Jiabin Zhang, Yonghui Zhang, Jun Chen, Fuquan Chen, Anfeng Xiao

**Affiliations:** 1Department of Bioengineering, Jimei University, Xiamen 361021, China; xiaoqiong129@jmu.edu.cn (Q.X.); 202012951069@jmu.edu.cn (X.W.); jiabin@jmu.edu.cn (J.Z.); yhz@jmu.edu.cn (Y.Z.); chenjun@jmu.edu.cn (J.C.); fqchenhy0109@jmu.edu.cn (F.C.); 2National R&D Center for Red Alga Processing Technology, Xiamen 361021, China; 3Fujian Provincial Engineering Technology Research Center of Marine Functional Food, Xiamen 361021, China; 4Xiamen Key Laboratory of Marine Functional Food, Xiamen 361021, China

**Keywords:** agar, extraction technology, pretreatment technique, quality change process

## Abstract

Optimizing the alkali treatment process alone without tracking the changes of algae and agar quality with each pretreatment process will not achieve the optimal agar yield and final quality. In this study, we monitored the changes of the morphology and weight of algae with each treatment process, and comprehensively analyzed the effects of each pretreatment process on the quality of agar by combining the changes of the physicochemical properties of agar. In conventional alkali-extraction technology, alkali treatment (7%, *w*/*v*) alone significantly reduced the weight of algae (52%), but hindered the dissolution of algae, resulting in a lower yield (4%). Acidification could solve the problem of algal hardening after alkali treatment to improve the yield (12%). In enzymatic extraction technology, agar with high purity cannot be obtained by enzyme treatment alone, but low gel strength (405 g/cm^2^) and high sulfate content (3.4%) can be obtained by subsequent acidification and bleaching. In enzyme-assisted extraction technology, enzyme damage to the surface fiber of algae promoted the penetration of low-concentration alkali (3%, *w*/*v*), which ensured a high desulfurization efficiency and a low gel degradation rate, thus improving yield (24.7%) and gel strength (706 g/cm^2^), which has the potential to replace the traditional alkali-extraction technology.

## 1. Introduction

Agar is one polysaccharide obtained from the cell wall of red seaweeds. It is commercially used as a thickening, gelling, stabilizing, and restructuring agents in various food formulations [1], as culture media in biological research [2], as a biodegradable carrier for drug delivery systems in the pharmaceutical industry [3], as well as agar-based nanoparticles in nano-bioengineering fields for wastewater treatment [4], as hydrogel films in functional food packaging, and wound dressing [5]. Agar is composed of two major components, agarose and agaropectin, respectively. Agarose is the gelling part of agar, consisting of repeating units of agarobiose by 1,3-linked-d-galactose and 1,4-linked-3,6-anhydro-l-galactose (3,6-AG) [6]. The hydroxyl groups of l-galactose can be substituted by sulfate, methoxyl, and pyruvate moieties, reducing the gelling ability of agar [7]. The industrial value of agar is mainly based on agar yield, gel strength, and purity rate. At present, the largest sources of agar worldwide are the genus *Gracilaria* and *Gelidium*, with only a small quantity coming from other species. In *Gelidium*, desulfation occurs as a natural internal transformation through an enzymatic process; therefore, extraction from *Gelidium* species results in high-quality agar, whereas, in *Gracilaria*, sulfate groups cannot be converted in the needed amount during the seaweed’s lifetime; therefore, desulfation should be promoted to an industrial level via a chemical method before agar extraction [8]. Although species of the genus *Gracilaria* generally produce agars with low gel strength, they are considered the most important source of commercially valuable agar for the food industry because they have been successfully cultivated in Chile and Indonesia, and the improved quality can be achieved via alkali treatment, which converts l-galactose-6-sulphate to 3,6-anhydro-l-galactose [8]. However, geographic factors, seasonal variations, growth stages, nutrient availability, and extraction process can influence the synthesis, yield, and chemistry of agar, thus leading to agar with a high heterogeneity [9]. Therefore, alkali treatment of *Gracilaria* species must be adapted for each species, and variables in the extraction process, such as temperature and alkali concentration, must be adjusted to achieve high efficiency desulfation, while avoiding yield losses during treatment.

Over the last decades, many researchers have studied alkaline treatment technology of different *Gracilaria* species. For example, Freile-Pelegrín et al. [10] found soaking of *Gracilaria cornea* with 3% or 5% NaOH rendered the agar with the highest gel strength (974–1758 g/cm^2^) and yield (14.5–22.1%). Praiboon et al. [11] found that agars extracted from *Gracilaria fisheri* and *Gracilaria edulis* pretreated with 5% NaOH showed a higher yield (34.3–39.6%) than the native agars (10.9–13.3%). Romero et al. [12] reported that treatment of alkali with 10% NaOH at 90 °C for 2 h was the optimal pretreatment process to achieve high-quality agar from *Gracilaria eucheumatoides*, with yield ranging from 22.9% to 29.0% and an average agar gel strength of 318 g/cm^2^. Yousefi et al. [13] reported that alkaline pretreatment of *Gracilaria corticata* with 5% NaOH rendered the agar with a prime quality (yield 31.48%; gel strength 364.6 g/cm^2^). Yarnpakdee et al. [14] found that the native agar from *Gracilaria tenuistipitata* without alkaline pretreatment had a yield of 17.1%. When alkaline pretreatment with NaOH and KOH at the concentration range of 3–7% were used for *Gracilaria tenuistipitata*, respectively, the high-quality agar with yield ranging from 23.6–26.1% was obtained, and gel strength increased by 77.5–80.4% (419–482 g/cm^2^) and 76.7–78.8% (406–446 g/cm^2^), respectively. Wang et al. [15] found that 6% alkali treatment at 80 °C for 1.5 h was the optimal pretreatment process for extracting high-quality agar from *Gracilaria tenuistipitata*; the process resulted in a gel strength of 1068.15 g/cm^2^ and high yield of 23.14%, which are higher than previous reports. However, in the current industrial practice of agar extraction, alkali treatment is only one of the many pre-treatment processes [8]. Other processes, such as acidification and bleaching, also have an important effect on the final quality of the product. Although many researchers have optimized the alkali process of agar extraction, most of them have only focused on the final agar obtained, and no in-depth study has been performed on the change of agar quality during extraction [10,11,12,13,14,15]. For example, a wide range of yield can be found in the literature due to different seaweeds and extraction techniques applied [10,11,12,13,14,15]. However, it was found that the yield of agar from different *Gracilaria* spp. is often more than 20%. According to a local agar-producing company (Greenfresh Food Co., Ltd., Zhangzhou, China, the enterprise with the highest production of agar in China), the agar yield from *Gracilaria lemaneiformis* ranged from 10.9% to 13.3% via the alkali method [16]. The agar yields reported by this company were low compared with those published in the literature [10,11,12,13,14,15]. As such, the reasons for the considerable differences in the final yield of agar from the actual content, the low yield of agar, and the specific processes that may cause agar loss remain unknown. Moreover, the alkali extraction of agar is not environmentally friendly. In achieving the green and sustainable upgrading of agar extraction, the effects of each process on the quality and yield of agar must be elucidated to realize the improvement of agar process [17]. Failure to protect the environment and human health and to maximize the agar yield and gel quality from limited natural seaweed resources may result in economic loss and threaten the sustainable supply of agar [18].

Therefore, this study aimed to analyze the change in agar quality in various procedures using *Gracilaria lemaneiformis* under different extraction technologies. The quality of agar from *Gracilaria lemaneiformis* was monitored in terms of yield, gel strength, sulfate and 3,6-AG contents, gel structure, and FTIR. Determining the effect of each process on agar quality may help improve the extraction process, enhance process efficiency, and obtain high-quality products.

## 2. Results and Discussion

### 2.1. Effects of the Procedures of Different Extraction Techniques on G. lemaneiformis

The conventional production of agar involves processes, including pretreatment, extraction, filtration, concentration, and dehydration. However, the pretreatment steps vary somewhat depending on the genus used. For *Gelidium*, the pretreatment step is comprised of a desulfuration treatment with a mild alkaline solution that can also eliminate phycoerythrine and prepare the seaweed for efficient extraction of agar with high quality. For *Gracilaria*, the alkali treatment before extraction is an essential procedure to enhance the gel strength. To extract agar, the seaweed is boiled in water that exceeds the boiling point. Acids often need to be carefully added, with pH adjusted to 6.3–6.5 to promote good extraction. Pressure extraction can shorten the processing time and increase agar yield. However, although acid boiling and pressure extraction can effectively promote the extraction of agar, both conditions can potentially destroy the quality of extracted agar. In addition, the agar dissolved in water must filter out the remaining algae, and the thermal filtrate should be cooled to form a gel. According to the required agar quality, the gel can be treated with sodium hypochlorite to reduce any color. After this treatment, the gel needs to be washed to remove bleach to prevent oxidative degradation of the gel. In general, the filtered agar solution contains about 1% agar, and the remaining 99% of the water must be removed from the gel by a freeze-thaw process or by the dewatering extrusion process. Therefore, the agar extraction process should be standardized for each species to maximize its yield and simultaneously obtain a good quality agar. The present study is the first attempt to clarify the effects of different pretreatment processes on the seaweed loss rate and agar yield, and to investigate the effects of pretreatment processes on the properties of agar. The specific experimental procedure is shown in Figure 1.

#### 2.1.1. Changes in *G. lemaneiformis* during Alkali Extraction

In alkali extraction of agar from *G. lemaneiformis*, algae were collected after sequentially treatment with alkali, acid, and bleaching treatments. The treated seaweeds were rinsed, freeze-dried, and photographed for analysis. The surface morphology of algae after freeze-drying is shown in Figure 2. The surface of algae without any treatment showed a thick cell wall composed of cellulose and presented thick folds with cracks after freeze drying (Figure 2A_2_,A_3_). After alkali treatment, alkali erosion destroys the algal epidermis and dissolves and destroys the cell wall of the finless porpoise, the cortex on the surface of algal becomes thinner, and the pith cells are broken or dissolved and become a disordered and loose incomplete cell structure. As a result, cracks form on the surface of algae and “white spots” are visible [19]. The thick folds on the surface of the finless porpoise are removed, revealing the uneven network with more gaps, which is due to the desulfuration caused by alkali treatment, resulting in the improved interconnection of polysaccharide chains (Figure 2B_2_,B_3_) [14]. Acid treatment further destroyed the cell wall, the cell membrane was dissolved, the cell wall of medulla was broken, and the alga was softened. This serious damage to the cell wall is beneficial for the dissolution of agar during hot water extraction, thereby increasing agar yield. As shown in Figure 2C_2_, the cell wall structure of alga was destroyed basically, and the reticular groove structure of agar on the surface of the algal body was also destroyed and presented a state of smooth concave uplift (Figure 2C_3_). During bleaching, the gelatinous layer on the surface of alga was further destroyed due to the acidic environment and strong oxidation of the bleaching solution; thus, the uneven surface of the algae gradually becomes smoother (Figure 2D_2_). Therefore, given the appearance of *G. lemaneiformis*, the alkali extraction reflected a process of gradual degradation of the algal cell wall and gradual exposure of colloid.

#### 2.1.2. Changes in *G. lemaneiformis* during Enzymatic-Extraction

In enzymatic extraction of agar from *G. lemaneiformis*, the surface morphology of alga after freeze drying is shown in Figure 3. The surface of alga without any treatment showed a thick layer of cellulose with cracks (Figure 3A). Cellulase acted on the algal cell wall and dissolved the cell wall gradually. After 2 h of cellulase treatment, the algal cell wall was destroyed, the transverse cracks of thick folds disappeared, and different degrees of depressions and irregular degradation marks were visible on the surface. A certain degree of pore structure was formed. The existence of pores indicates that the fiber layer on the surface of alga is hydrolyzed by cellulase (Figure 3B_2_,B_3_). After acid treatment, the cell wall was softened and thinned again, showing a honeycomb structure and resulting in an increase in the loss rate of alga. The honeycomb structure is the embodiment of agar gel structure caused by water sublimation after vacuum freeze drying (Figure 3C_3_). After bleaching, the cell walls on the surface of the honeycomb layer were degraded again, and gelatinous dissolution occurred in some places (Figure 3D_2_,D_3_).

#### 2.1.3. Changes in *G. lemaneiformis* in Enzyme-Assisted Alkali Extraction

After enzyme, alkali, acid, and bleaching treatments, the representative alga after each step of rinsing was freeze-dried and photographed for analysis. As shown in Figure 4, after 1 h of enzyme treatment, the thick folds on the surface of alga became thin and rough (Figure 4B_2_,B_3_). After 3% alkali treatment, the algal surface changed from rough to smooth, and the pores in the reticular structure disappeared. This phenomenon might be due to the alkali treatment after enzyme treatment that made it easy to peel off the algae that was further desulfurized by alkali [16]. Some honeycomb structures were observed in the algal body, but the pore structure was still covered by the cell wall and did not show protrusion after water sublimation (Figure 4C_2_,C_3_). This finding indicated that the cell wall from the algal surface was not completely removed, although most of the cellulose from the algal cell wall could be removed by alkali with low content after enzymatic treatment. After acid treatment, the lamellar hemicellulose cell wall connected to the algae was removed, and the algal surface became smoother (Figure 4D_2_,D_3_). After bleaching, the algal surface had a protuberant honeycomb structure, but colloid leaching phenomenon was not observed. Therefore, as indicated by the appearance of *G. lemaneiformis*, the enzyme-assisted alkali extraction was milder than alkaline extraction, reflecting the mild degradation of cellulose without exposure of colloid.

### 2.2. Effects of Extraction Technologies on Physicochemical Properties of Agar

#### 2.2.1. Effect of Pretreatment Procedure on Sulfate Content, 3,6-AG Content, Gel Strength, Algae Loss Rate, and Agar Yield

As shown in Figure 5A_2_, comparing with all agar samples from *G. lemaneiformis* for the gel strength, the agar from untreated *G. lemaneiformis* had a lower gel strength. The gelation of agar is formed by the aggregation of polysaccharide chains through the helical conformation of hydrogen bonds. The presence of charged groups (sulfate groups) may interfere with the intermolecular hydrogen bonds formed by the double helix. Alkaline pretreatment contributed to chain straightening and regularity in the polymer by cleavage of the C-6 sulfate group and closure of the ring to form the 3,6-AG [20]. Thus, gel strength increased by 152.5% (301–760 g/cm^2^) when NaOH with a concentration of 7% was used for pretreatment. Although the 3,6-AG content of agar extracted after alkali treatment was ca. 44%, which was similar to the agar extracted after acid treatment and bleaching treatment, the gel strength of agar extracted after alkali treatment was lower than that of the agar extracted after acid and bleaching treatments. By contrast, the loss rate of algae after alkali treatment reached 53% and was mainly due to alkali treatment that removes most of the impurities, such as protein, pigment, and cellulose, and the loss of gelatinous substances in the cell wall of the algae with the alkali, resulting in a remarkable loss of algal mass. The loss rate of algae gradually increased with the further degradation of the fibers in the algae and the loss of part of the exposed colloids after acid treatment and bleaching. However, after alkali treatment of algae, the agar yield was only 5%, which was mainly because alkali treatment not only increased gel strength but also hardened the algal body. As such, the agar in algal body was difficult to dissolve and extract in hot water. In addition, the damaged cellulose in the algal body was mixed with agar solution in hot water and was easy to be filtered out together with the gel solution during filtration. This process leads to impure agar, resulting in low gel strength after alkali treatment. After acid treatment, the algal body softened after further degradation of surface cellulose, and the softened body could be dissolved during hot water extraction. It was easy to filter, so the agar yield after acid treatment was high. As shown in Figure 5A_2_, bleaching treatment could further remove algal pigment and decrease algal mass but had no limited effect on agar yield.

During enzyme extraction of agar from *G. lemaneiformis* (Figure 5B_1_,B_2_), the sulfate and 3,6-AG contents of agar extracted from seaweed subjected to enzyme, acid, and bleaching treatments had no significant difference from those of agar extracted from untreated seaweed. Thus, the majority of the native polysaccharide in red algae is present as L-galactose sulfate, which does not form a gel. Although enzyme treatment could not convert L-galactose sulfate to 3,6-AG and increase the gel strength of the resultant agar, cellulase could act on the algal cell wall, promoting the dissolution of agar during hot water extraction. In addition, after acid and bleaching treatments, the loss rates of algae were 19% and 23%, respectively, which were 42% and 52% higher than those after enzyme treatment, indicating that a large amount of cellulose, pigment, and protein still existed in algae after enzyme treatment. This result also corresponded to the low whiteness and transparency of agar after enzymatic treatment. Although the loss rate of algae increased gradually, the yield of agar only changed slightly, which indicated that there was no phenomenon of agar running out after different processing procedures. From the gel strength, compared with the agar extract by untreated seaweed, the agar extracted after enzyme and bleaching treatments had a high gel strength, whereas the gel strength of the agar extracted after acid treatment was low. This phenomenon is mainly due to the algal body that easily softens after acid treatment. During hot water extraction, fiber disperses in agar solution and with the agar solution through the filter cloth together. This result is consistent with a marked decrease in transparency as indicated in Table 1. At present, the sulfated polysaccharides obtained from various seaweeds have attracted great interest due to their unique biological effects, such as anticancer, antioxidant, antifungal, antiviral, and anticoagulant activities [21,22]. For instance, Liu et al. [23] found that the sulfated polysaccharide from *G. lemaneiformis* exhibited the ability to promote the function of regulatory Treg cells and alleviate allergy symptoms, which may be developed as a functional food component for the allergic patients. Cui et al. [24] reported that the sulfated polysaccharides extracted from red algae have effective anti-inflammatory activities and protected THP-1 cells against LPS-induced toxicity. Therefore, natural sulfated agar is no doubt a kind of sulfated polysaccharide resource that has the potential to nutraceutical or pharmaceutical application. However, both alkali extraction and enzymatic-assisted extraction can greatly reduce the sulfate content of agar, and direct hot water extraction cannot get high purity agar sulfated polysaccharide, so enzymatic extraction is a more suitable method to obtain agar sulfated polysaccharide.

After enzyme treatment, the contents of sulfate and 3,6-AG and gel strength of agar were not significantly different from those extracted from untreated *G. lemaneiformis* (Figure 5C_1_,C_2_). However, the weight of seaweed decreased by 10%, and the agar yield was reduced by 9%. The loss weight of seaweed was mainly attributed to the degradation of cellulose. The decrease in the agar extraction rate was due to the damaged algal cell wall conducive to the dissolution of agar during hot water extraction. As such, the extracted agar contained less impurities. However, agar from untreated *G. lemaneiformis* was difficult to dissolve after hot water extraction despite the softening of algae. During the filtration stage, the agar should be filtered out by extrusion. Then, a large amount of cellulose passed through the press cloth, resulting in the large number of impurities in agar that led to high yields. Native agar from *G. lemaneiformis* had 3.8% (*w/w*) sulfate ester; a strong reduction was detected in the extract obtained after 3% NaOH treatment, and no further decrease was observed after acid and bleaching treatments. Pretreatment with enzyme before alkaline treatment can destroy the cell walls of *G. lemaneiformis*, promote the penetration and absorption of alkali, and further cleave sulfate ester at C-6 of l-galactose with a small amount of alkali. After alkali treatment, the sulfate content of agar decreased, the 3,6-AG content increased, and the gel strength of agar increased, but the loss rate of algae also increased sharply to 36.8%, which was mainly caused by the degradation of cellulose and the loss of pigment and protein. With the subsequent seaweed treatment, such as acid and bleaching treatments, the loss rate of algae was further increased. However, the yield and gel strength of agar were higher than those of agar after alkali treatment, mainly because the algae hardened after alkali treatment, which was not conducive to the dissolution of agar and subsequent filtration. After acid treatment, the further degradation of cellulose softened the algae, facilitating the dissolution of agar and obtaining an agar solution with improved purity after filtration. Therefore, the yield and gel strength increased after acid treatment. Algal loss rate increased after bleaching treatment, suggesting that bleaching treatment can effectively remove algal pigments, such as chlorophyll and phycobilin, corresponding to its improved whiteness and transparency.

As shown in Figure 5D_1_, an inverse correlation between the sulfate and 3,6-AG contents of agar was observed. The native agar exhibited the lowest 3,6-AG content (33.9%) and, therefore, the highest sulfate content (3.8%). By contrast, alkali-treated samples showed much higher 3,6-AG contents (42%). Among these samples, the difference in the 3,6-AG content (~40%) between alkaline extraction and enzyme-assisted extraction was not significant. A slightly higher 3,6-AG content and similar fraction of sulfate was observed for agar obtained via enzyme extraction. The differences in 3,6-AG content may be due to the extraction procedure in the samples. The decrease in agar yield in all samples may be related to the degradation of polysaccharides during treatment and agar loss by diffusion during processing. Native agar from *G. lemaneiformis* had 38% (*w/w*) of yields (Figure 5D_2_); a strong reduction was detected in the extract obtained after alkali treatment, and no further decrease was observed after enzyme extraction. Although the gel strength of agar obtained enzyme-assisted extraction was slightly lower than that of agar via alkaline extraction, the yield of agar extracted enzyme-assisted extraction of agar was significantly higher than that of agar obtained via alkaline extraction, indicating that enzyme-assisted pretreatment could enhance the desulfurization efficiency of alkali and prevent agar degradation by high-strength alkali.

#### 2.2.2. Effect of Pretreatment Procedures on the Whiteness, Transparency, Viscosity, Dissolving Temperature, Gelling Temperature, and Melting Temperature of Agar

Differences in color were observed among agars and *G. tenuistipitata* pretreated via different procedures at various extraction processes. Native agar (without pretreatment) had a brown yellowish color. Pigments, such as chlorophyll, carotenoid, and phycoerythrobilin, in algae, leached out during extraction most likely due to the dark color of native agars [25]. During alkaline pretreatment, algal cells are destroyed, and the pigments in the cells are easy to undergo photolysis, which can remove part of the pigments, which is actually phycobilin because phycohemoglobin and phycocyanin are water-soluble pigments that can be dissolved in dilute alkali solution. Li et al. [25] noted that phycoerythrobilin was removed from phycoerythrin during the alkaline pretreatment of *Gracilaria lemaneiformis*. As a result, those pigments could be released from cells with ease during alkaline pretreatment. With NaOH pretreatment (7%, *w*/*v*) in the alkaline process, the obtained agar had improved whiteness values compared with those pretreated with enzyme and NaOH (3%, *w*/*v*) during enzyme extraction and enzyme-assisted extraction, respectively. This result reconfirmed the effectiveness of alkali in destroying or removing pigments in algae, in which the color of agar could be further improved. However, alkali treatment did not completely remove the pigment from the algae, as can be seen from the darkening of the algae after acid treatment. This may be due to the fibers inside the algae that were further destroyed after acid treatment, and the pigment inside the alga migrated outward under the action of osmotic pressure. However, it could not be completely dissolved due to the small pore size of the outer fiber network and deposited on the subsurface of the alga. After hot water extraction of the acid-treated alga, the alga was dissolved and the pigment was released, resulting in a darker color. When NaClO was used for treatment, the increase in whiteness values was more pronounced, the resulting agar exhibited a slight creamy yellowish color. Thus, the alkaline solution at high concentrations was able to remove most pigments from the algal cell.

Transparency can be used to reflect the gel properties and purity of agar. As shown in Table 1, a lower transparency was obtained in gels of agar extracted from algae with NaOH pretreatment during alkali extraction. The lower transparency was mainly due to agar impurity. The hardened algae after alkali treatment were difficult to dissolve, and a large number of impurities passed through the press cloth with the agar solution during extrusion and filtration, making the agar solution more opaque. Removing impurities during the agar dehydration was also difficult. This phenomenon is more obvious during enzymatic extraction than in other methods because enzyme and acid treatments can only destroy the fiber structure of the algal surface. During hot water immersion, many insoluble fibers and pigments are dispersed in the agar solution and filtered out together with the agar during filtration, resulting in uneven turbidity and the dark color of the agar solution. However, in the enzyme-assisted extraction, agar transparency increased gradually, mainly because the algae treated with low concentration of alkali had moderate hardness and were easy to extract by hot water. Then, the algal fibers were degraded, and the pigment was removed gradually after acid and bleaching treatments. The resulting agar gel had improved transparency with no impurities.

Variations in the viscosity of agars with different treatments were observed (Table 1). All agars with alkaline pretreatment had a higher viscosity (12.4 cP, 3%; 19.5 cP, 7%) than native agar (9.3 cP). The contamination of non-gelling components in native agar may be one of the reasons for the low viscosity [11]. For example, during enzymatic extraction, the agar viscosity extracted from *G. tenuistipitata* after acid treatment was the lowest, mainly because the fibers on the surface of algal could not be degraded by cellulase and acid. After hot water extraction, a large number of fibers were dispersed in the agar solution and filtered out with the agar in the process of pressure filtration. In addition, the presence of sulfate groups in the structure might lead to electrostatic repulsion between the polysaccharide chains. The reduced entanglement of the polysaccharide chain possibly resulted in the decreased viscosity. After alkaline pretreatment, the degree of chain entanglement enhanced with the increase of the 3,6-AG content, thus leading to the higher viscosity. Moreover, the viscosity of agar solution also depends on the molecular weight or the chain length of molecules [26]. Yarnpakdee et al. [14] found that viscosity of agar from *G. tenuistipitata* pretreated with NaOH at all concentrations used was higher than those pretreated with KOH because the polysaccharide chain could be further degraded by KOH. Meanwhile, agar extracted from *G. tenuistipitata* with high alkaline concentration (7% NaOH, *w*/*v*) showed a decreased viscosity. Similarly, Praiboon et al. [11] reported that the decrease in viscosity of agar pretreated with 5% NaOH was due to the hydrolysis of the polysaccharide chain by alkaline. In general, the viscosity could reflect the purity or quality of the product. In production, the viscosity of agar could also be adjusted by different pretreatment methods according to the actual needs.

Dissolving, gelling, and melting temperatures of agars with different treatments are presented in Table 1. The native agar from *G. tenuistipitata* showed lower gelling and melting temperatures of 30.4 °C and 82.8 °C, respectively. It was found that native agars from *G. corticata* from collected from Kenya, *G. tenuistipitata* collected from the Philippines, and *G. blodgettii* and *G. crassissima* collected from the Yucatan Peninsula showed different gelling and melting temperatures, and their gelling and melting temperatures are within the range of 42–45 °C and 86–88 °C, respectively [27,28,29]. The gelling and melting temperatures vary depending on the area where the algae are harvested. In addition, both gelling and melting temperatures are affected by different extraction processes. For agar with alkaline pretreatment during alkali extraction (7% NaOH, *w*/*v*) and enzyme-assisted extraction (3% NaOH, *w*/*v*), those with NaOH pretreatment formed gel at temperature of 42.4 °C and 34.8 °C, respectively, and the corresponding gels molt at 86.4 °C and 84.4 °C. That is, agar with higher gelling and melting temperatures can be extracted from *G. tenuistipitata* after alkaline pretreatment. The high melting temperature is attributed to the high energy required to cleave the network, indicating that the alkali pretreated agar gels were more stable than native agar. The high melting temperature is ascribed to the high energy required to break down the network, indicating that the agar gel with alkaline pretreatment were more stable than native agar [30]. Villanueva et al. [30] reported that the gelling and melting temperatures of native agar from *G. vermiculophylla* were the 21.6–26.4 °C and 62.7–70.0 °C, respectively; however, both gelling and melting temperatures rose to 31.0–35.8 °C and 73.6–80.4 °C, respectively, when *G. vermiculophylla* was treated with 4% NaOH. In addition, the substitution degree of methoxy group on the agar polysaccharide chain was also one of important factors affecting the gelling temperature. The amount and location of sulfate groups were also thought to inhibit or delay the gelation mechanism [11]. Although the methoxyl content of agar from *G. tenuistipitata* was not measured in this study, the increase in gelling temperature and melting temperatures in agar from *G. tenuistipitata* could be due to the increased amount of methoxyl group and decreased sulfate substitution in the agar, respectively. In addition, the melting temperature of agar was positively correlated with its molecular weight. A higher molecular weight of agar can increase the stable interactions between the agar polymer structures and require higher temperature to melt the agar gel [31]. For example, the melting temperature of agar via enzyme extraction was lower (82.2 °C ± 0.4°C) than that of native agar (82.8 °C), which is probably due to the influence of molecular weight. The melting temperature of agar was affected by the pretreatment procedures used in different extraction techniques, but within the range specified by the United States Pharmacopeia (80–85 °C); this indicated possible applications in the food and pharmaceutical industries, especially for products that require sterilization [32]. Therefore, during agar extraction, different procedures are related to the physical and chemical properties of agar, and how to coordinate the relationship between the procedures and the quality of agar is the guarantee to obtain the best technology.

### 2.3. FT-IR Analysis of Agar Extracted from Each Procedure

The agars were analyzed by FTIR to better understand the results related to extraction procedure. Figure 6 shows the FTIR spectra of agars extracted from each procedure derived from different extraction technologies. The band near 3500 cm^−1^ is attributed to the O–H stretching and deformation band. The band at 2932 cm^−1^ indicates the C-H stretching band. The weak peak at 2888 cm^−1^ indicates the very low methylation degree (O–CH_3_) of the agar obtained (Figure 6B,C). The band at 1607 cm^−1^, assigned to amide I vibrations, indicates that proteins are present (Figure 6B) [33]. The presence of proteins in agar indicates that the pigment is not completely removed [1]. The bands at 1380 and 1250 cm^−1^ are related to the total sulfate content and indicate agar quality because the presence of α-l-galactose 6-sulfate units decreases the gel strength. As expected, native agar had the largest band at 1250 cm^−1^, after alkaline extraction (Figure 6A), the absorption peak of the sulfate group in the bleaching stage is the smallest, which is also related to the highest strength of agar gel after bleaching as shown in Figure 6A. In enzymatic extraction (Figure 6B), the characteristic absorption peak strength at 1380 cm^−1^ of the agar after enzyme treatment, acid treatment, and bleaching treatment at 1380 cm^−1^ was the same, suggesting that these pretreatment processes in enzymatic extraction are not helpful for the removal of the sulfate group. The broad absorption region between 1100 and 1000 cm^−1^, as well as the band at 1150 cm^−1^ are attributed to the skeleton of galactans [34]. The band at 930 cm^−1^ is related to the C–O–C group of 3,6-AG [21]. As shown in Figure 6D, the band intensity was increased near 930 cm^−1^ by alkaline pretreatment, especially by 7% NaOH pretreatment. This is mainly due to the conversion of sulfate substitution of 6-sulphate-α-l-galactose to 3,6-AG by alkali pretreatment. The results were consistent with the content of 3,6-AG determined by 7% NaOH (alkali extraction) and 3% NaOH (enzyme assisted alkali extraction) pretreatment with agar colorimetry (Figure 5D_1_). The band at 890 cm^−1^ is attributed to the C–H bending at the anomeric carbon in galactans’ residues [35]. The absorption at 850, 820 and 805 cm^−1^ indicated the presence of sulfate groups on the C-4 and C-6 of G moieties and C2 in LA moieties, respectively [21]. After alkali extraction (7% NaOH, *w*/*v*) and enzyme-assisted (3% NaOH, *w*/*v*) agar extraction, weak bands appeared at 1250 cm^−1^, and no bands appeared at 805 and 820 cm^−1^, indicating that alkaline and enzyme-assisted extraction have the same effect on eliminating unstable sulfate groups (C2 in LA and C6 in G moieties) [16]. In addition, the peak at 850 cm^−1^ representing the sulfate group at C-4 was present in all samples after alkaline pretreatment, indicating that the axial sulfate group at C-4 is alkaline-stable sulfate groups. Overall, alkaline treatment had a great effect on the structure and functional group of agar, while acid treatment, bleaching treatment, and high temperature extraction had little effect on the structure of agar.

## 3. Materials and Methods

### 3.1. Chemicals

Dried *G. lemaneiformis* was obtained from Greenfresh (Fujian) Food Stuff Co., Ltd. (Zhangzhou, China). Neutral cellulase (activity: 15,000 U/g) was purchased from Novozymes (China) Biotechnology Co., Ltd. (Tianjin, China). Resorcinol, potassium sulfate, barium chloride, sodium hydroxide, sodium hypochlorite, hydrochloric acid, oxalate, sulfuric acid, fructose, and gelatin were purchased from Sinopharm Chemical Reagent Co., Ltd. (Shanghai, China) All chemicals received were of analytical grade and used without further purification.

### 3.2. Agar Extraction from G. lemaneiformis

The collected *G. lemaneiformis* samples with impurities such as salt, soil, and shells were washed to remove impurities, dried, and set aside.

#### 3.2.1. Traditional Alkali Extraction

The seaweeds (30 g) were first soaked in sodium hydroxide solutions (7% *w*/*v*, V_NaOH_:W_seaweeds_ = 20:1) for 3 h at 90 °C [16]. After alkali treatment, the seaweeds were then washed and soaked with water until neutral. Then, the seaweeds were acidified in two steps: first, they were initially acidified with sulfuric acid solution (0.064%, *w*/*v*) for 40 min and with an acid solution composed of sulfuric acid (0.064%, *w*/*v*), oxalate (0.052%, *w*/*v*), and EDTA (0.012%, *w*/*v*) for another 40 min. The acid-treated seaweeds were then washed and soaked with water until neutral. The next major step was the bleaching processes. The seaweeds were soaked in sodium hypochlorite solution (0.1%, *w*/*v*) for 40 min, washed, and soaked with water until neutral. Finally, the seaweeds (V_water_:W_seaweeds_ = 20:1) were heated to 95–102 °C until the seaweeds were completely dissolved. The seaweed extracts were then pressure filtered, dehydrated, and dried. The specific process is shown in Figure 1. Samples were obtained after performing each procedure (alkali treatment, acidification, and bleaching) and rinsing. Then, the samples were dried in a 50 °C blast dryer to constant weight, and weighed to determine the loss rate of seaweed, agar yield, and other physicochemical properties. In addition, the treated seaweed samples collected were freeze-dried and scanned by an electron microscope to observe the changes in seaweed after each procedure.

#### 3.2.2. Enzyme-Assisted Extraction of Agar

The seaweeds (30 g) were first soaked in cellulase solutions (4 U/mL, V_cellulase_:W_seaweeds_ = 20:1) at 50 °C for 1 h. After enzyme treatment, the seaweeds were treated with sodium hydroxide solution (3% *w*/*v*, V_NaOH_:W_seaweeds_ = 20:1) at 87 °C for 3 h. The seaweeds were then washed and soaked with water until neutral pH. The seaweeds were acidified in one step, i.e., the seaweeds were impregnated in acid solution composed of sulfuric acid (0.016%, *w*/*v*), oxalate (0.016%, *w*/*v*), and EDTA (0.012%, *w*/*v*) for 20 min. The acid-treated seaweeds were then washed and soaked with water until neutral pH. Thereafter, the seaweeds were soaked in sodium hypochlorite solution (0.06%, *w*/*v*) for 20 min and then washed and soaked with water until neutral. Finally, the seaweeds (V_water_:W_seaweeds_ = 20:1) were heated to 95–102 °C until the seaweeds were completely dissolved. The seaweed extracts were then pressure filtered, dehydrated, and dried.

#### 3.2.3. Enzymatic-Extraction of Agar

The seaweeds (30 g) were first soaked in cellulase solutions (8 U/mL, V_cellulase_:W_seaweeds_ = 20:1) at 50 °C for 3 h. After enzyme treatment, the seaweeds were acidified in one step, that is, the seaweeds were impregnated in acid solution composed of sulfuric acid (0.05%, *w*/*v*), oxalate (0.05%, *w*/*v*), and EDTA (0.012%, *w*/*v*) for 40 min. The acid-treated seaweeds were then washed and soaked with water until neutral. After that, the seaweeds were soaked in sodium hypochlorite solution (0.25%, *w*/*v*) for 20 min and then washed and soaked with water until neutral. Finally, the seaweeds (V_water_:W_seaweeds_ = 20:1) were heated to 95–102 °C until the seaweeds were completely dissolved. The seaweed extracts were then pressure filtered, dehydrated, and dried.

### 3.3. Characterization of Treated Seaweed and Extracted Agar

#### 3.3.1. Scanning Electron Microscopy (SEM)

*G. lemaneiformis* samples collected after each process were subjected to vacuum freeze drying (Telstar, LyoQuest-85, Terrassa, Spain) for approximately 24 h. The morphology of samples was then analyzed by SEM (Hitachi, S-4800, Tokyo, Japan).

#### 3.3.2. Fourier Transform Infrared Spectroscopy (FT-IR)

The agar samples were blended with KBr powder and pressed into thin slices. The FT-IR spectrum of samples was recorded by using a FT-IR spectrophotometer (Thermo Fisher, Nicolet iS50, Waltham, MA, USA) in a wavelength range from 4000 to 500 cm^−1^.

#### 3.3.3. Determination of Physicochemical Properties

The sulfate content of agar samples was measured turbidimetrically using BaCl_2_-gelatin method after hydrolysis in 0.5 M HCl as described by Yarnpakdee et al. [14]. First, a 0.5% gelatin solution was prepared and placed in a 4 °C refrigerator overnight. Subsequently, 1% BaCl_2_ was added to the solution, mixed thoroughly, and left to stand for several hours. Approximately 0.1 g of agar samples was transferred in a colorimetric tube, and 25 mL of 1 M HCl was added. The colorimetric tube was placed in a water bath at 100 °C and digested for 5 h. After cooling the tube to room temperature, activated carbon was added for decolorization of the sample, and the digestive fluid was filtered. K_2_SO_4_ was dried to a constant weight at 105 °C. Approximately 0.1088 g of K_2_SO_4_ was accurately weighed, and dissolved with 100 mL of 1 M HCl. The standard curve was drawn with 1 mL of different concentrations of K_2_SO_4_ standard solution mixed with 3 mL of gelatin-BaCl_2_ solution. The absorbance was measured at 360 nm after blending for 10 min. Finally, the absorbance of the sample was measured at 360 nm, and the sulfate content was calculated using the standard curve.

3,6-AG content was determined colorimetrically using the resorcinol-acetal method as described by Yaphe et al. [36]. First, 1.5 mg/mL resorcinol solution was prepared, and 0.04% (*v/v*) 1,1-acetal solution was stored at 4 °C in the refrigerator. Approximately 9 mL of resorcinol solution, 1 mL of 1,1- diethoxyethane solution, and 100 mL of 12 M concentrated HCl were mixed into the solution before analysis. Subsequently, 1 mL of the sample solution was extracted and placed in an ice bath for 5 min, and 5 mL of resorcinol reagent was sufficiently mixed into the sample solution. The mixture was placed in a water bath at 80 °C for 15 min, transferred in an ice bath for 1.5 min, and measured at a wavelength of 554 nm. Finally, the 3,6-anhydro-L-galactose content was calculated using the fructose standard curve.

Gel strength of agar samples (1.5%, *w*/*v*) was determined using methods described by Lee et al. [37]. A 1.5% (*w*/*v*) agar solution was prepared and heated until fully dissolved. The gel strength was determined by pouring the solution into a Petri dish and setting it aside overnight at 20 °C. The gel strength was measured within 20 s and calculated as gram per square centimeter.

Melting and gelling temperature of agar samples (1.5%, *w*/*v*) were analyzed using methods described by Freile-Pelegrin et al. [27]. Melting temperature of the gel in test tubes was measured by placing a glass bead (5 mm diameter) on the gel surface. The test tube rack with test tube was transferred to the water bath at boiling temperature. The melting temperature was recorded with a digital thermometer when the bead sank into the solution. The same test tubes were kept at room temperature to measure the gelling temperature. The tubes were tilted up and down in a water bath at room temperature until the glass bead ceased moving. The gel temperature in the tube was immediately measured by introducing a digital thermometer into the agar gel.

The dissolving temperature was measured as described by Cao et al. [38]). In a thermostatic water bath, agar (1.5 g) and deionized water (98.5 g) were stirred in a 250 mL four-necked flask equipped with a mechanical stirrer, a reflux condenser, and a temperature controller. The heating rate was uniform in all cases at 1 C/min, and the dissolving temperatures were recorded by monitoring the temperature at which the agar was fully dissolved in water.

Transparency of agar gel (1.5%, *w*/*v*) was determined using methods described by Normand et al. [39]. Agar was dissolved in boiling deionized water to obtain a final concentration of 1.5% (*w*/*v*). The sample solution (1%, *w*/*v*) was placed in the colorimetric ware and then incubated at 20 °C for 12 h. The transparency of agar gel was measured by transmittance (%) at 700 nm with distilled water as a blank.

Apparent viscosity of agar samples (1.5%, *w*/*v*) was measured at 80 °C using a viscometer (Brookfield, DV-C, Middleboro, MA, USA).

Whiteness of agar was determined by whiteness analyzer (Xinrui Instruments, WSB-2, Shanghai, China) after passing through 80 mesh sieves.

The yields of agars were calculated based on the dry weight of the initial seaweed.

### 3.4. Statistical Analysis

All experiments were carried out in triplicate, and the average was calculated. Data were analyzed for variance and expressed as mean ± standard deviation. Duncan’s multipolar test was used to compare the mean values. SPSS 17.0 for Windows was used to analyze all the data.

## 4. Conclusions

Traditional extraction methods have been widely studied and commercially employed despite their limitations. Understanding the effects of each process on the quality and yield of agar is the premise of improving the agar extraction process. The results showed that alkali treatment alone significantly reduced the weight of algae but hindered the dissolution of algae, resulting in a lower yield. Acidification could solve the problem of algal hardening after alkali treatment to improve the yield. Agar with high purity cannot be obtained by enzyme treatment alone, but low gel strength and high sulfate content can be obtained by subsequent acidification and bleaching. Enzyme treatment damage to the surface fiber of algae promoted the penetration of low-concentration alkali, which ensured a high desulfurization efficiency and a low gel degradation rate, thus improving yield and gel strength, which has the potential to replace the traditional alkali-extraction technology. These findings indicate that the optimization of a single procedure is not enough to improve agar quality. Only the perfect cooperation of each process can extract agar products that meet the quality requirements.

## Figures and Tables

**Figure 1 marinedrugs-19-00617-f001:**
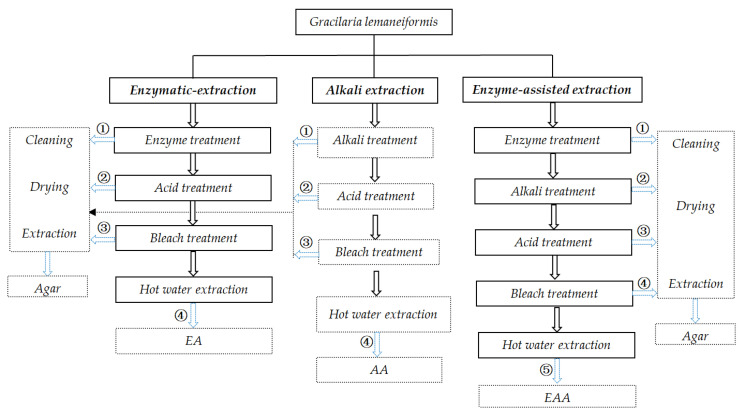
Experimental scheme for agar extraction. Note: numeric character represents agar obtained from algae treated by various processes.

**Figure 2 marinedrugs-19-00617-f002:**
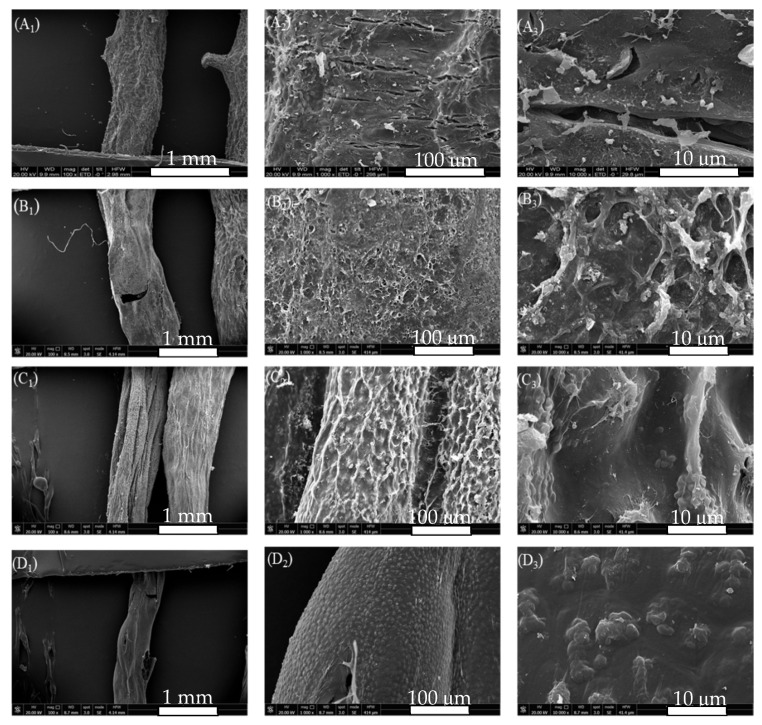
SEM of *G. lemaneiformis* during alkali extraction, (**A**) surface structure of untreated *G. lemaneiformis*; (**B**) surface structure of *G. lemaneiformis* after alkali treatment; (**C**) surface structure of *G. lemaneiformis* after acid treatment; (**D**) surface structure of *G. lemaneiformis* after bleaching treatment. Scale bar in photos represent 1 mm (left), 100 μm (center), and10 μm (right), respectively.

**Figure 3 marinedrugs-19-00617-f003:**
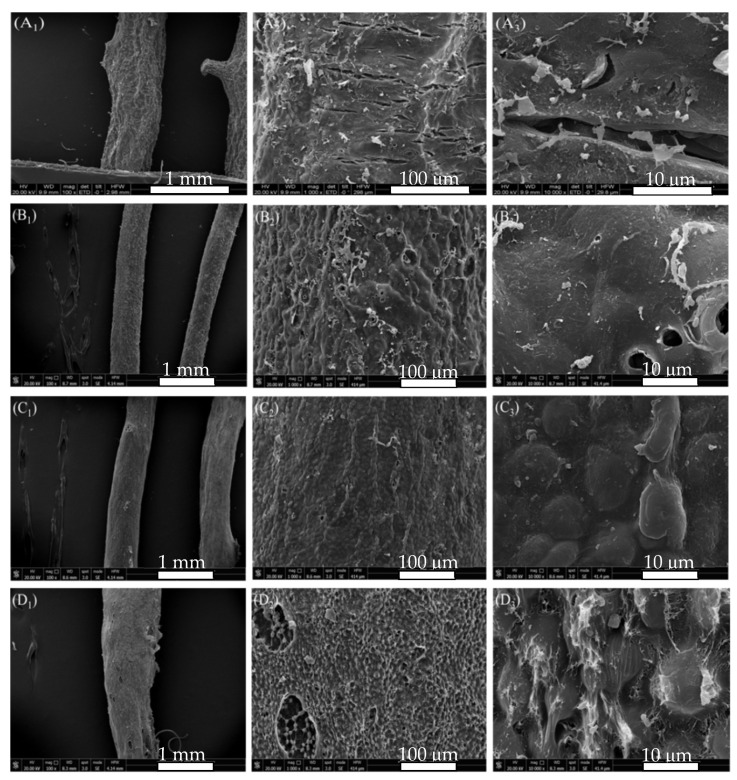
SEM of *G. lemaneiformis* during enzymatic-extraction process. (**A**) surface structure of untreated *G. lemaneiformis*; (**B**) surface structure of *G. lemaneiformis* after enzyme treatment; (**C**) surface structure of *G. lemaneiformis* after acid treatment; (**D**) surface structure of *G. lemaneiformis* after bleaching treatment. Scale bar in photos represent 1 mm (left), 100 μm (center), and10 μm (right), respectively.

**Figure 4 marinedrugs-19-00617-f004:**
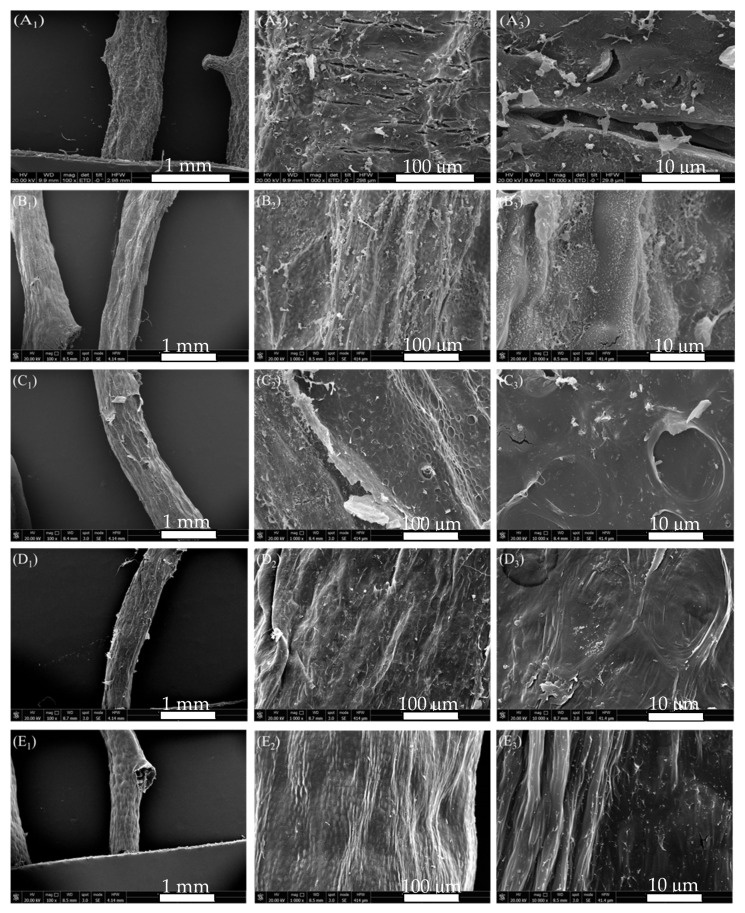
SEM of *G. lemaneiformis* during enzyme-assisted extraction. (**A**) surface structure of untreated *G. lemaneiformis*; (**B**) surface structure of *G. lemaneiformis* after enzyme treatment; (**C**) surface structure of *G. lemaneiformis* after alkali treatment; (**D**) surface structure of *G. lemaneiformis* after acid treatment; (**E**) surface structure of *G. lemaneiformis* after bleaching treatment. Scale bar in photos represent 1 mm (left), 100 μm (center), and10 μm (right), respectively.

**Figure 5 marinedrugs-19-00617-f005:**
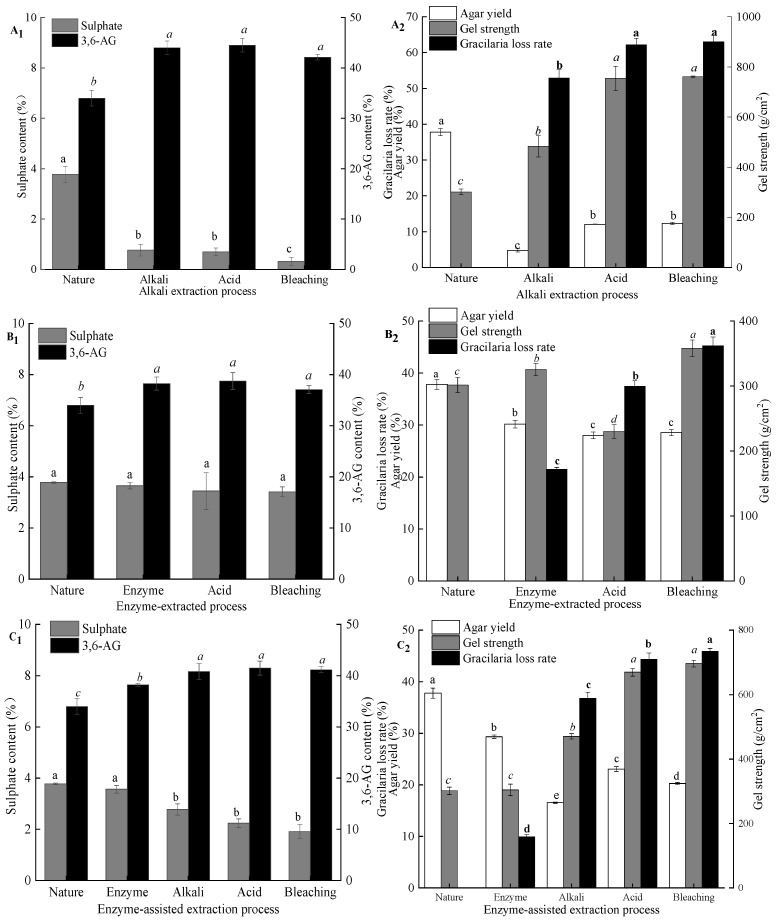
Effect of pretreatment techniques and green extraction technologies on the physicochemical properties of agar. (**A_1_**,**A_2_**) alkali extraction process; (**B_1_**,**B_2_**) enzymatic-extraction process; (**C_1_**,**C_2_**) enzyme-assisted alkali extraction process; (**D_1_**,**D_2_**) comparison of three extraction technologies; nature: a sample extracted without pretreatment; note: different lowercase superscripts within the same column indicate the significant differences (*p* < 0.05).

**Figure 6 marinedrugs-19-00617-f006:**
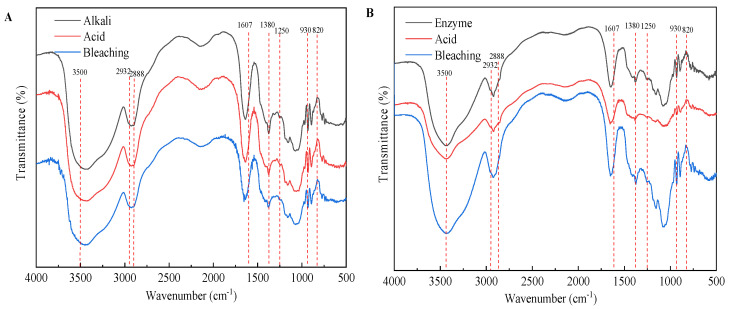
FTIR of agar from different extraction processes. (**A**) alkali extraction; (**B**) enzymatic-extraction; (**C**) enzyme-assisted extraction; (**D**) comparison of three extraction technologies.

**Table 1 marinedrugs-19-00617-t001:** Effect of different extraction processes on physicochemical properties of agar.

PP	Nature	Alkali Extraction	Enzymatic-Extraction	Enzyme-Assisted Extraction
	Alkali	Acid	Bleaching	Extraction	Enzyme	Acid	Bleaching	Extraction	Enzyme	Alkali	Acid	Bleaching	Extraction
WH (%)	35.6 ± 0.3	45.7 ± 0.2 ^b^	43.1 ± 1.1 ^c^	61.3 ± 0.5 ^a^	62.9 ± 0.5 ^a^	42.9 ± 0.8 ^b^	32.9 ± 0.9 ^c^	46.8 ± 0.5 ^a^	44.4 ± 0.4 ^c^	36.5 ± 0.4 ^b^	34.3 ± 0.1 ^c^	34.8 ± 0.2 ^b^	42.4 ± 0.3 ^a^	50.2 ± 0.6 ^b^
GC	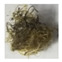	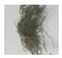	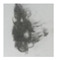	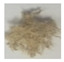	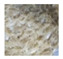	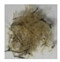	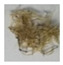	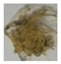	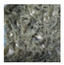	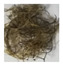	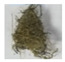	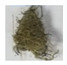	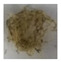	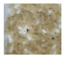
AC	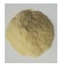	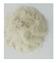	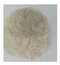	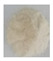	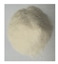	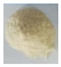	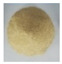	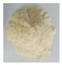	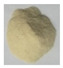	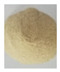	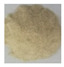	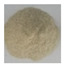	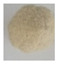	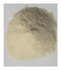
TR (%)	49.8 ± 0.5	44.4 ± 0.5 ^c^	54.8 ± 0.8 ^b^	59.9 ± 0.8 ^a^	62.0 ± 0.2 ^a^	52.6 ± 0.5 ^b^	34.6 ± 0.3 ^c^	59.0 ± 1.2 ^a^	58.0 ± 0.3 ^b^	52.9 ± 0.9 ^d^	56.3 ± 0.4 ^c^	57.6 ± 0.4 ^b^	60.8 ± 0.6 ^a^	61.9 ± 0.1 ^a^
VI (cP)	9.3 ± 0.2	19.5 ± 0.6 ^c^	28.1 ± 1.6 ^a^	24.5 ± 1.1 ^b^	20.0 ± 0.4 ^a^	12.5 ± 0.0 ^b^	3.8 ± 0.2 ^c^	18.2 ± 0.8 ^a^	18.8 ± 0.3 ^b^	8.7 ± 1.1 ^c^	12.4 ± 0.4 ^b^	13.6 ± 0.9 ^b^	17.7 ± 0.6 ^a^	18.1 ± 0.5 ^b^
DT (°C)	93.1 ± 1.0	91.1 ± 0.1 ^b^	92.4 ± 1.2 ^ab^	93.1 ± 0.1 ^a^	89.2 ± 0.0 ^b^	86.3 ± 0.0 ^b^	84.9 ± 0.6 ^c^	87.2 ± 0.0 ^a^	87.4 ± 1.0 ^c^	85.9 ± 0.2 ^b^	87.1 ± 0.5 ^a^	86.4 ± 0.0 ^b^	87.4 ± 0.1 ^a^	89.3 ± 0.1 ^b^
MT (°C)	82.8 ± 0.1	86.4 ± 0.1 ^b^	89.6 ± 0.2 ^a^	89.5 ± 0.6 ^a^	89.5 ± 0.0 ^a^	82.9 ± 0.1 ^a^	71.9 ± 0.1 ^b^	82.5 ± 0.5 ^a^	82.2 ± 0.4 ^a^	79.2 ± 0.1 ^c^	84.4 ± 0.1 ^b^	84.5 ± 0.0 ^b^	85.5 ± 0.2 ^a^	85.9 ± 0.1 ^a^
GT (°C)	34.0 ± 0.2	42.4 ± 0.4 ^a^	41.6 ± 1.0 ^b^	41.1 ± 0.3 ^ab^	36.0 ± 0.7 ^b^	35.1 ± 0.2 ^b^	32.9 ± 0.5 ^c^	36.9 ± 0.2 ^a^	32.8 ± 0.5 ^c^	33.6 ± 0.2 ^c^	34.8 ± 0.3 ^b^	33.9 ± 0.1 ^c^	35.8 ± 0.4 ^a^	35.8 ± 0.5 ^b^

Note: Different lowercase superscripts within the same column indicate the significant differences (*p* < 0.05). **PP**: physicochemical property; **WH**: whiteness; **GA**: *G. tenuistipitata* color; **AC**: agar color; **VI**: viscosity; **TR**: transparency; **DT**: dissolving temperature; **MT**: melting temperature; **GT**: gelling temperature. **Nature**: a sample extracted without pretreatment.

## Data Availability

Data is contained within the article.

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
