# Peer review of "Pretreatment Techniques and Green Extraction Technologies for Agar from Gracilaria lemaneiformis"

_marinedrugs, 2021, doi:10.3390/md19110617_

Round 1

Reviewer 1 Report

  1. The authors have not mentioned anywhere the need for such a study and its novelty.
  2. Line 231: change G. lemaneiformis into italics
  3. Line 465: change G. lemaneiformis into italics
  4. Line 469: vNaOH:wseaweeds font size
  5. Section 3.2.1 Please add reference for traditional alkali treatment
  6. Line 477: vwater:wseaweeds font size
  7. Line 486: vcellulase:wseaweeds font size
  8. Line 486: at what pH was the extraction conducted? Was a buffer used? If yes, what kind of buffer was used?
  9. Line 490: Not sure if you can impregnate seaweed. Can you change to ‘suspended in’ instead please?
  10. The only difference between section 3.2.2 and 3.2.3 seems to be an additional line which explains the soaking of seaweed in sodium hypochlorite. Perhaps this section can be combined to explain the difference in the procedures since the authors have used the same sentences to explain both processes. Also, the term ‘enzyme extraction’ is misleading since no enzymes are extracted from the seaweed. Please rename the extraction process and rephrase the section.
  11. A flow diagram of the enzyme assisted extraction process would be beneficial.
  12. There are two ‘Fig 1’ in the manuscript.
  13. Please remove the ‘Fig 1’ in the methodology section. This figure gives the reader no details on the extraction processes discussed in the paper.
  14. Figure 4 shows results of the effect of pretreatment techniques on the physicochemical properties of agar. The authors have done statistical analysis to reveal that the effects are significantly difference from each other. Please incorporate the kind of statistical analysis conducted and the software (with version) used for it in the methodology section.  

Author Response

  1. Reviewers' comments:

Comment 1: The authors have not mentioned anywhere the need for such a study and its novelty.

Answer: Thank you for your comment. We have expounded the need and novelty of this article in the preface. As follows,

Although many researchers have optimized the alkali process of agar extraction, most of them have only focused on the final agar obtained, and no in-depth study has been performed on the change of agar quality during extraction. For example, a wide range of yield can be found in the literature due to different seaweeds and extraction techniques applied. However, it was found that the yield of agar from different Gracilaria spp. is often more than 20%. According to a local agar-producing company (Greenfresh Food Co., Ltd., the enterprise with the highest production of agar in China), the agar yield from Gracilaria lemaneiformis ranged from 10.9% to 13.3% via the alkali method. The agar yields reported by this company were low compared with those published in the literature. As such, the reasons for the considerable differences in the final yield of agar from the actual content, the low yield of agar, and the specific processes that may cause agar loss remain unknown. Moreover, the alkali extraction of agar is not environmentally friendly. In achieving the green and sustainable upgrading of agar extraction, the effects of each process on the quality and yield of agar must be elucidated to realize the improvement of agar process [16]. Failure to protect the environment and human health and to maximize the agar yield and gel quality from limited natural seaweed resources may result in economic loss and threaten the sustainable supply of agar [17,18].

Therefore, this study aimed to analyze the change in agar quality in various procedures using Gracilaria lemaneiformis under different extraction technologies. The quality of agar from Gracilaria lemaneiformis was monitored in terms of yield, gel strength, sulfate and 3,6-AG contents, gel structure, and FTIR. Determining the effect of each process on agar quality may help improve the extraction process, enhance process efficiency, and obtain high-quality products.”

In general, in current studies, researchers mostly focused on optimizing the extraction process of agar, unfortunately, the changes of algae after various treatments were not studied in detail, which would affect the effect of subsequent treatments. Meanwhile, the changes in the properties of agar inside algae before and after various treatments were also not studied in detail. Therefore, our optimized process does not allow the front-line production personnel to understand the nature of the production process, and the existing optimization process is difficult to be applied to practical production, and the current research is difficult to solve the problems encountered in practical production. By studying the changes of algae and agar inside algae after various treatments, we can find the source of problems and provide solutions when the quality of agar products changes in actual production. This is the need and novelty of this study.

Comment 2: Line 231: change G. lemaneiformis into italics

Answer: Thank you for your suggestion. G. lemaneiformis has changed into G. lemaneiformis.

Comment 3: Line 465: change G. lemaneiformis into italics

Answer: Thank you for your suggestion. G. lemaneiformis has changed into G. lemaneiformis.

Comment 4: Line 469: vNaOH:wseaweeds font size

Answer: Thank you for your suggestion. vNaOH:wseaweeds has changed into VNaOH:Wseaweeds.

Comment 5: Section 3.2.1 Please add reference for traditional alkali treatment

Answer: Thank you for your suggestion. We have added reference for traditional alkali treatment in this article.

Comment 6: Line 477: vwater:wseaweeds font size

Answer: Thank you for your suggestion. vNaOH:wseaweeds has changed into VNaOH:Wseaweeds.

Comment 7: Line 486: vcellulase:wseaweeds font size

Answer: vNaOH:wseaweeds has changed into VNaOH:Wseaweeds.

Comment 8: Line 486: at what pH was the extraction conducted? Was a buffer used? If yes, what kind of buffer was used?

Answer: To simulate the actual production of agar, neutral cellulase was used and reacted in tap water with no buffer prepared.

Comment 9: Line 490: Not sure if you can impregnate seaweed. Can you change to ‘suspended in’ instead please?

Answer: Thank you for your suggestion. The sentence “...the seaweeds were impregnated in acid solution…” has been changed to ‘...the seaweeds were suspended in acid solution…’.

Comment 10: The only difference between section 3.2.2 and 3.2.3 seems to be an additional line which explains the soaking of seaweed in sodium hypochlorite. Perhaps this section can be combined to explain the difference in the procedures since the authors have used the same sentences to explain both processes. Also, the term ‘enzyme extraction’ is misleading since no enzymes are extracted from the seaweed. Please rename the extraction process and rephrase the section.

Answer: The main difference is that the enzyme-assisted extraction method treats algae with enzymes and then 3% sodium hydroxide, while the enzyme-assisted extraction method does not treat algae with alkali. The quality of agar extracted by enzyme-assisted extraction method is similar to that extracted by traditional alkali method (NaOH, 7% w/v). Besides, the term ‘enzyme extraction’ has been changed to ‘enzymatic-extraction’.

As shown below,

“3.2.2. Enzyme-assisted extraction of agar

The seaweeds (30 g) were first soaked in cellulase solutions (4 U/mL, Vcellulase:Wseaweeds=20:1) at 50 °C for 1 h. After enzyme treatment, the seaweeds were treated with sodium hydroxide solution (3% w/v, VNaOH:Wseaweeds=20:1) at 87 °C for 3 h. The seaweeds were then washed and soaked with water until neutral pH. The seaweeds were acidified in one step, i.e. the seaweeds were impregnated in acid solution composed of sulfuric acid (0.016%, w/v), oxalate (0.016%, w/v) and EDTA (0.012%, w/v) for 20 min. The acid-treated seaweeds were then washed and soaked with water until neutral pH. Thereafter, the seaweeds were soaked in sodium hypochlorite solution (0.06%, w/v) for 20 min and then washed and soaked with water until neutral. Finally, the seaweeds (Vwater:Wseaweeds=20:1) were heated to 95 °C–102 °C until the seaweeds were completely dissolved. The seaweed extracts were then pressure filtered, dehydrated, and dried.

3.2.3. Enzymatic-extraction of agar

The seaweeds (30 g) were first soaked in cellulase solutions (8 U/mL, Vcellulase:Wseaweeds=20:1) at 50 °C for 3 h. After enzyme treatment, the seaweeds were acidified in one step, that is, the seaweeds were impregnated in acid solution composed of sulfuric acid (0.05%, w/v), oxalate (0.05%, w/v), and EDTA (0.012%, w/v) for 40 min. The acid-treated seaweeds were then washed and soaked with water until neutral. After that, the seaweeds were soaked in sodium hypochlorite solution (0.25%, w/v) for 20 min and then washed and soaked with water until neutral. Finally, the seaweeds (Vwater:Wseaweeds=20:1) were heated to 95 °C–102 °C until the seaweeds were completely dissolved. The seaweed extracts were then pressure filtered, dehydrated, and dried.”

Comment 11: A flow diagram of the enzyme assisted extraction process would be beneficial.

Answer: The flow diagram of the three different extraction processes was shown in Fig. 1.

Comment 12: There are two ‘Fig 1’ in the manuscript.

Answer: This error has been corrected in the article

Comment 13: Please remove the ‘Fig 1’ in the methodology section. This figure gives the reader no details on the extraction processes discussed in the paper.

Answer: The flow chart helps the reader understand how the sample is prepared, so we want to keep it.

Figure 1. Experimental scheme for agar extraction. Note: numeric character represents agar obtained from algae treated by various processes.

Comment 14: Figure 4 shows results of the effect of pretreatment techniques on the physicochemical properties of agar. The authors have done statistical analysis to reveal that the effects are significantly difference from each other. Please incorporate the kind of statistical analysis conducted and the software (with version) used for it in the methodology section.

Answer: Thank you for your suggestion. We refined this in the methods section, as shown below,

“3.5. Statistical analysis

All experiments were carried out in triplicate, and the average was calculated. Data were analyzed for variance and expressed as mean ± standard deviation. Duncan’s multipolar test was used to compare the mean values. SPSS 17.0 for Windows was used to analyze all the data.”

Reviewer 2 Report

It is a interesting article based in the chemical and extraction optimization of agar, however, have various notorious major questions:

Abstract: please first sentences need to be completed and rewritten, to the reader understand What and how the problem is important 

Introduction, general lack of bibliographic support for large part of the sentences

Results and discussion needs to be separated (analyze of results first, figures or tables after, and discussion of results after).

Methods can be better explained and structured, section 3.4

But overall, the manuscript without a part about medical or pharmaceutical application, the manuscript is without the Marine drugs scope of publication... there is aneed to add more bibliography and work baut the potential to nutraceutical or pharmaceutical application of the agar extracted...

See annexed for specific questions in the manuscript

Author Response

Reviewers' comments:

It is a interesting article based in the chemical and extraction optimization of agar, however, have various notorious major questions:

Comment 1: Abstract: please first sentences need to be completed and rewritten, to the reader understand What and how the problem is important.

Answer: Thank you for your suggestion. We have reorganized the abstract to make them more specific. As shown below,

Abstract: Optimizing the alkali treatment process alone without tracking the changes of algae and agar quality with each pretreatment process will not achieve the optimal agar yield and final quality. In this study, we monitored the changes of the morphology and weight of algae with each treatment process, and comprehensively analyzed the effects of each pretreatment process on the quality of agar by combining the changes of the physicochemical properties of agar. In conventional alkali-extraction technology, alkali treatment (7%, w/v) alone significantly reduced the weight of algae (52%), but hindered the dissolution of algae, resulting in a lower yield (4%). Acidification could solve the problem of algal hardening after alkali treatment to improve the yield (12%). In enzymatic extraction technology, agar with high purity cannot be obtained by enzyme treatment alone, but low gel strength (405 g/cm2) and high sulfate content (3.4%) can be obtained by subsequent acidification and bleaching. In enzyme-assisted extraction technology, enzyme damage to the surface fiber of algae promoted the penetration of low-concentration alkali (3%, w/v), which ensured a high desulfurization efficiency and a low gel degradation rate, thus improving yield (24.7%) and gel strength (706 g/cm2), which has the potential to replace the traditional alkali-extraction technology.”

Comment 2: Introduction, general lack of bibliographic support for large part of the sentences

Answer: Thank you for your suggestion. We have supplemented the relevant literature in the manuscript.

Comment 3: Results and discussion needs to be separated (analyze of results first, figures or tables after, and discussion of results after).

Answer: The journal does not strictly require the results to be separated from the discussion. In addition, since the content of the article involves a variety of comparative analysis, the conclusion and discussion together seem to better illustrate the essence of the problem. We hope you can agree with us to keep the original format of the article.

Comment 4: Methods can be better explained and structured, section 3.4

Answer: Thank you for your suggestion. Section 3.4 has been better explained and structured, as shown below.

“The sulfate content of agar samples was measured turbidimetrically using BaCl2-gelatin method after hydrolysis in 0.5 M HCl as described by Yarnpakdee et al. [14]. First, 0.5% gelatin solution was prepared and placed in a 4 °C refrigerator overnight. Subsequently, 1% BaCl2 was added to the solution, mixed thoroughly, and left to stand for several hours. Approximately 0.1 g of agar samples was transferred in a colorimetric tube, and 25 mL of 1 M HCl was added. The colorimetric tube was placed in a water bath at 100 °C and digested for 5 h. After cooling the tube to room temperature, activated carbon was added for decolorization of the sample, and the digestive fluid was filtered. K2SO4 was dried to a constant weight at 105 °C. Approximately 0.1088 g of K2SO4 was accurately weighed, and dissolved with 100 mL of 1 M HCl. The standard curve was drawn with 1 mL of different concentrations of K2SO4 standard solution mixed with 3 mL of gelatin-BaCl2 solution. The absorbance was measured at 360 nm after blending for 10 min. Finally, the absorbance of the sample was measured at 360 nm, and the sulfate content was calculated using the standard curve.

3,6-AG content was determined colorimetrically using the resorcinol-acetal method as described by Yaphe et al. [33]. First, 1.5 mg/mL resorcinol solution was prepared, and 0.04% (v/v) 1,1-acetal solution was stored at 4 °C in the refrigerator. Approximately 9 mL of resorcinol solution, 1 mL of 1,1- diethoxyethane solution, and 100 mL of 12 M concentrated HCl were mixed into the solution before analysis. Subsequently, 1 mL of the sample solution was extracted and placed in an ice bath for 5 min, and 5 mL of resorcinol reagent was sufficiently mixed into the sample solution. The mixture was placed in a water bath at 80 °C for 15 min, transferred in an ice bath for 1.5 min, and measured at a wavelength of 554 nm. Finally, the 3,6-anhydro-L-galactose content was calculated using the fructose standard curve.

Gel strength of agar samples (1.5%, w/v) was determined using methods described by Lee et al. [34]. A 1.5% (w/v) agar solution was prepared and heated until fully dissolved. The gel strength was determined by pouring the solution into a Petri dish and setting it aside overnight at 20 °C. The gel strength was measured within 20 s and calculated as gram per square centimeter.

Melting and gelling temperature of agar samples (1.5%, w/v) were analyzed using methods described by Freile-Pelegrin et al. [23]. Melting temperature of the gel in test tubes was measured by placing a glass bead (5 mm diameter) on the gel surface. The test tube rack with test tube was transferred to the water bath at boiling temperature. The melting temperature was recorded with a digital thermometer when the bead sank into the solution. Same test tubes were kept at room temperature to measure the gelling temperature. The tubes were tilted up and down in a water bath at room temperature until the glass bead ceased moving. The gel temperature in the tube was immediately measured by introducing a digital thermometer into the agar gel.

The dissolving temperature was measured as described by Cao et al. [35]). In a thermostatic water bath, agar (1.5 g) and deionized water (98.5 g) were stirred in a 250 ml four-necked flask equipped with a mechanical stirrer, a reflux condenser and a temperature controller. The heating rate was uniform in all cases at 1 ºC/min, and the dissolving temperatures were recorded by monitoring the temperature at which the agar was fully dissolved in water.

Transparency of agar gel (1.5%, w/v) was determined using methods described by Normand et al. [36]. Agar were dissolved in boiling deionized water to obtain a final concentration of 1.5% (w/v). The sample solution (1%, w/v) was placed in the colorimetric ware and then incubated at 20 °C for 12 h. The transparency of agarose gel was measured by transmittance (%) at 700 nm with distilled water as a blank.”

Comment 5: But overall, the manuscript without a part about medical or pharmaceutical application, the manuscript is without the Marine drugs scope of publication... there is a need to add more bibliography and work baut the potential to nutraceutical or pharmaceutical application of the agar extracted...

Answer: Thank you for your suggestion. We add more bibliography to expound the potential application to nutraceutical or pharmaceutical of the agar extracted. As shown Section 2.2.1,

“At present, the sulfated polysaccharides obtained from various seaweeds have attracted great interest due to their unique biological effects, such as anticancer, antioxidant, antifungal, antiviral, and anticoagulant activities [21,22]. For instance, Liu et al. [23] found that the sulfated polysaccharide from G. lemaneiformis exhibited the ability to promote the function of regulatory Treg cells and alleviate allergy symptoms, which may be developed as a functional food component for the allergic patients. Cui et al. [24] reported that the sulfated polysaccharides extracted from red algae have effective anti-inflammatory activities and protected THP-1 cells against LPS-induced toxicity. Therefore, natural sulfated agar is no doubt a kind of sulfated polysaccharide resource that have the potential to nutraceutical or pharmaceutical application. However, both alkali extraction and enzymatic-assisted extraction can greatly reduce the sulfate content of agar, and direct hot water extraction can not get high purity agar sulfated polysaccharide, so enzymatic extraction is a more suitable method to obtain agar sulfated polysaccharide.”

Comment 6: See annexed for specific questions in the manuscript

Answer: Thank you for your suggestion. These specific questions have been revised and highlighted in manuscript.

Reviewer 3 Report

In their study, Xiao and colleagues evaluated different pretreatment techniques to extract agar and characterize agar quality. Overall, this is a well-designed study; however, the following minor issues need to be addressed. 

  1. Abstract and conclusion do not reflect the actual findings of the study. Particularly, there are no data that describe agar quality as affected by different pretreatment techniques. Even, the conclusion is missing in abstract, indicating which pretreatment technique produces a better result.
  2. In Figure 4B2, it seems that there is no difference in agar yield between acid and bleaching treatment, but these carry a different significant value (c and b). Figure 4D1 and D2 are wrongly indicated in the figure. However, authors did not mention in the figure legend what different letters in the bar graph mean.
  3. Table 1 is hard to read.
  4. Other minor issues:

Line 64: Gracilariaedulis (Gracilaria edulis)

Line 195-196: In general, the gel formed……………. This sentence should be rewritten.

Line203-204, Line 231, Line 252 and elsewhere: G. lemaneiformis (should be etalic)

Line 281: agars (agar)

Line 434: Figure 5(D) (Fig. 5D)

Line 439: Fig. D1 (Fig. 4D)

Line 531: Chronologically Figure 1 should be Figure 6

Author Response

Reviewers' comments:

In their study, Xiao and colleagues evaluated different pretreatment techniques to extract agar and characterize agar quality. Overall, this is a well-designed study; however, the following minor issues need to be addressed.

Comment 1: Abstract and conclusion do not reflect the actual findings of the study. Particularly, there are no data that describe agar quality as affected by different pretreatment techniques. Even, the conclusion is missing in abstract, indicating which pretreatment technique produces a better result.

Answer: Thank you for your comment. We have reorganized the abstract and conclusion to make them more specific, as shown below.

Abstract: Optimizing the alkali treatment process alone without tracking the changes of algae and agar quality with each pretreatment process will not achieve the optimal agar yield and final quality. In this study, we monitored the changes of the morphology and weight of algae with each treatment process, and comprehensively analyzed the effects of each pretreatment process on the quality of agar by combining the changes of the physicochemical properties of agar. In conventional alkali-extraction technology, alkali treatment (7%, w/v) alone significantly reduced the weight of algae (52%), but hindered the dissolution of algae, resulting in a lower yield (4%). Acidification could solve the problem of algal hardening after alkali treatment to improve the yield (12%). In enzymatic extraction technology, agar with high purity cannot be obtained by enzyme treatment alone, but low gel strength (405 g/cm2) and high sulfate content (3.4%) can be obtained by subsequent acidification and bleaching. In enzyme-assisted extraction technology, enzyme damage to the surface fiber of algae promoted the penetration of low-concentration alkali (3%, w/v), which ensured a high desulfurization efficiency and a low gel degradation rate, thus improving yield (24.7%) and gel strength (706 g/cm2), which has the potential to replace the traditional alkali-extraction technology.”

4. Conclusions

Traditional extraction methods have been widely studied and commercially employed despite their limitations. Understanding the effects of each process on the quality and yield of agar is the premise of improving agar extraction process. The results showed that alkali treatment alone significantly reduced the weight of algae, but hindered the dissolution of algae, resulting in a lower yield. Acidification could solve the problem of algal hardening after alkali treatment to improve the yield. agar with high purity cannot be obtained by enzyme treatment alone, but low gel strength and high sulfate content can be obtained by subsequent acidification and bleaching. Enzyme treatment damage to the surface fiber of algae promoted the penetration of low-concentration alkali, which ensured a high desulfurization efficiency and a low gel degradation rate, thus improving yield and gel strength, which has the potential to replace the traditional alkali-extraction technology. These findings indicate that the optimization of a single procedure is not enough to improve agar quality. Only the perfect cooperation of each process can extract agar products that meet the quality requirements.”

Comment 2: In Figure 4B2, it seems that there is no difference in agar yield between acid and bleaching treatment, but these carry a different significant value (c and b). Figure 4D1 and D2 are wrongly indicated in the figure. However, authors did not mention in the figure legend what different letters in the bar graph mean.

Answer: Thank you for your comment. We reviewed the article carefully again and corrected the problems mentioned above.

As shown below,

Figure 5. Effect of pretreatment techniques and green extraction technologies on the physicochemical properties of agar. (A1, A2) alkali extraction process; (B1, B2) enzyme extraction process; (C1, C2) enzyme assisted alkali extraction process; (D1, D2) Comparison of three extraction technologies; Nature: a sample extracted without pretreatment; Note: Different lowercase superscripts within the same column indicate the significant differences (P < 0.05).

Comment 3: Table 1 is hard to read.

Answer: Table 1 mainly compares the physicochemical properties of agar extracted from seaweed by various processes. The process flow chart is adjusted to Figure 1 for better understanding.

Figure 1. Experimental scheme for agar extraction. Note: numeric character represents agar obtained from algae treated by various processes.

Comment 4: Line 64: Gracilariaedulis (Gracilaria edulis)

Answer: Thank you for your comment. Gracilariaedulis has changed into Gracilaria edulis.

Comment 5: Line 195-196: In general, the gel formed……………. This sentence should be rewritten.

Answer: Thank you for your suggestion. The sentence “In general, the gel formed after filtering is contained only 1% agar, the remaining 99% of the water has to be removed from the gel, either through a freeze-thaw process or by squeezing it out under pressure.” has been changed to “In general, the filtered agar solution contains about 1% agar, and the remaining 99% of the water must be removed from the gel by a freeze-thaw process or by dewatering extrusion process.”

Comment 6: Line203-204, Line 231, Line 252 and elsewhere: G. lemaneiformis (should be etalic)

Answer: Thank you for your comment. G. lemaneiformis has changed into G. lemaneiformis.

Comment 7: Line 281: agars (agar)

Answer: Thank you for your comment. This error has been corrected in the manuscript. As shown below,

“As shown in Fig. 4, an inverse correlation between the sulfate and 3,6-AG contents of agar was observed.”

Comment 8: Line 434: Figure 5(D) (Fig. 5D)

Answer: Thank you for your comment. This error has been corrected in the manuscript. As shown below,

“As shown in Fig. 5D, the band intensity was increased near 930 cm-1 by alkaline pretreatment, especially by 7% NaOH pretreatment.”

Comment 9: Line 439: Fig. D1 (Fig. 4D)

Answer: Thank you for your comment. This error has been corrected in the manuscript. As shown below,

“The results were consistent with the content of 3, 6-AG determined by 7% NaOH (alkali extraction) and 3% NaOH (enzyme assisted alkali extraction) pretreatment with agar colorimetry (Fig. 4D1).”

Comment 10: Line 531: Chronologically Figure 1 should be Figure 6.

Answer: Thank you for your comment. This error has been corrected in the manuscript.

Round 2

Reviewer 1 Report

Can you reduce the font size of table 1 so that the numbers fall in a single line? Or perhaps you can split the table in to two or more. In its current format, the table looks very cluttered.